# Stripping Model for Short GRBs: The Impact of Nuclear Data

**Andrey Yudin** [1,*], **Nikita Kramarev** [1,2], **Igor Panov** [1,3] and **Anton Ignatovskiy** [1,3]

1   National Research Center Kurchatov Institute, pl. Kurchatova 1, Moscow 123182, Russia; kramarev-nikita@mail.ru (N.K.); panov_iv@itep.ru (I.P.); lirts@phystech.edu (A.I.)

2   Sternberg Astronomical Institute, Lomonosov Moscow State University, Universitetsky pr. 13, Moscow 119234, Russia

3   Moscow Institute of Physics and Technology, Landau Phystech School of Physics and Research, Moscow 141701, Russia

\*   Correspondence: yudin@itep.ru

**Abstract:** We investigate the impact of forthcoming nuclear data on the predictions of the neutron star (NS) stripping model for short gamma-ray bursts. The main area to which we pay attention is the NS crust. We show that the uncertain properties of the NS equation of state can significantly influence the stripping time $t_{\rm str}$, the main dynamical parameter of the model. Based on the known time delay ($t_{\rm str} \approx 1.7$ s) between the peak of the gravitational wave signal GW170817 and the detection of gamma photons from GRB170817A, we obtain new restrictions on the nuclear matter parameters, in particular, the symmetry energy slope parameter: $L < 114.5$ MeV. In addition, we study the process of nucleosynthesis in the outer and inner crusts of a low-mass NS. We show that the nucleosynthesis is strongly influenced by both the forthcoming nuclear data and the equation of state of the NS matter.

**Keywords:** gamma-ray bursts; neutron stars; stripping model; r-process





## 1. Introduction

The last stages of the evolution of compact binary star systems attract great attention of astrophysicists as sources of powerful multi-messenger transients. In particular, it has long been suggested that the binary neutron star (NS) inspirals are sources of short gamma-ray bursts (GRBs) and gravitational wave (GW) events [1–4]. It was also assumed [5] that the NS-NS coalescences should produce supernova-like combined optical, ultraviolet, and infrared transients, later called kilonovae [6]. But only in 2017, with the joint registration of the GW signal GW170817 and the gamma-ray burst GRB170817A [7,8], as well as the subsequent detection of the kilonova AT2017gfo [9], did these theoretical predictions receive reliable observational confirmation [10].

The conventional picture of what happens in the last seconds of the NS-NS binary evolution in the most general terms can be described as follows (see, e.g., [11,12]): two NSs approach each other due to the energy and angular momentum loss to emit GW radiation and finally merge, forming a supramassive NS or a black hole (BH). The neutron-rich matter ejected during the NS coalescence is an ideal place for the rapid neutron-capture process (or r-process) and the subsequent formation of heavy elements, accompanied by kilonova emission (see [13] for a review and references). At the same time, some part of matter is ejected in the form of narrow collimated relativistic streams (or jets) responsible for the observed short GRB (see [4,14–16] for a detailed description). The scenario described above will be hereinafter referred to as the NS merging scenario.

The alternative to it is the stripping mechanism [1,2]. In this model, the NSs, at the moment of their closest approach (on the order of tens of kilometers), instead of merging begin to exchange mass, where the massive component tears off (or strips off) matter from the low-mass one. The latter reaches the minimum NS mass [17]

and explodes, producing a spherically symmetric GRB [18] and contributing to the cosmic synthesis of heavy elements [19,20]. A more detailed description of the stripping model, as well as a comparison of its predictions with observational data on the above-mentioned peculiar gamma-ray burst GRB170817A, can be found in [21,22]. In particular, we have pointed out that the time delay between the GW170817 peak and the GRB170817A registration [8], in the framework of the stripping model, corresponds to the duration of the stable mass transfer.

An important issue is the conditions under which the mechanism of stripping rather than merging is implemented. Our preliminary calculations [23] show that this condition weakly depends on the total mass of the system and is determined mainly by the mass ratio of the components. The stripping model is realized at $M_2/M_1 \lesssim 0.8$, so about a quarter of the observed galactic NS-NS binaries [24] with known masses must finish their evolution in accordance with this scenario. In this context, it is also important to mention the recent discovery [25] of very low-mass NS ($0.77 M_\odot$ approximately). In any case, this issue requires further careful study.

In this article, we focus on how the nuclear data uncertainties influence the predictions of the stripping model. We discuss variations in the NS equation of state (EoS), which are predicted by various models uncertainties in the nuclear composition of the NS crust and differences in the results of nucleosynthesis calculations related to the ambiguity of the cross-sections used. Moreover, we mainly discuss the region of thermodynamic parameters characteristic of the inner and outer NS crusts, that is, the region of sub-nuclear densities. The reason for that becomes clear from an inspection of Figure 1.

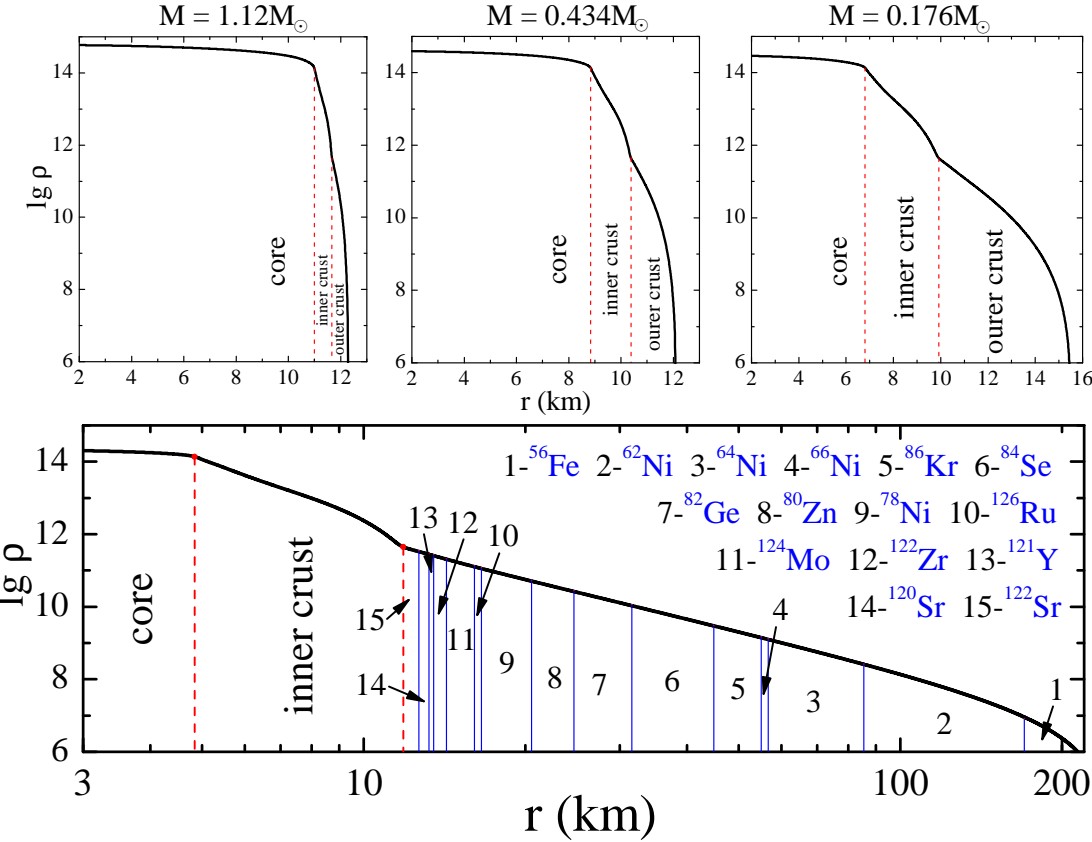

**Figure 1.** Structure of NSs with different masses. The lower wide panel corresponds to the minimum NS mass. The regions of the core, inner, and outer crusts are identified. See text for details.

The four panels of this figure show the dependencies of the density $\rho$ in the NS as a function of the radial coordinate $r$ for the BSk25 EoS [26]. The NS mass values for each of the upper panels are shown at the top; the lower panel corresponds to the minimum

NS mass (about 0.085 $M_\odot$). In addition, on the same panel, numbers with labels show the nuclear composition of the outer crust for this EoS (see also Table 1 below). The boundaries of the core, inner, and outer crusts are shown by vertical red dashed lines. In the NS with moderate mass ($M = 1.12\ M_\odot$), the outer and inner crusts occupy only a small fraction of the star's volume. However, as the NS mass decreases, this fraction sharply increases and the core size decreases. In the NS of minimum mass, the crust extends over more than 200 km. It is in the crust region that a shock wave is generated producing gamma radiation in the stripping model [18,27]; here, active nucleosynthesis (r-process) occurs accompanying the explosive destruction of the low-mass NS (LMNS) [19]. This is why the crust region is so important for the stripping model.

**Table 1.** Trajectory parameters for calculating nucleosynthesis in layers of the outer crust.

| № * | Composition | | $T_9^{\mathrm{max}}(t)$ | | $\lg \rho^{\mathrm{max}}(t)$ | | $Y_e$ | | $\Delta M, 10^{-4} M_\odot$ | | $\sum M, 10^{-4} M_\odot$ | |
|---|---|---|---|---|---|---|---|---|---|---|---|---|
| | BSk22 | BSk25 | BSk22 | BSk25 | BSk22 | BSk25 | BSk22 | BSk25 | BSk22 | BSk25 | BSk22 | BSk25 |
| 18 | $^{128}$Sr | - | 0.98 | - | 11.64 | - | 0.297 | - | 0.88 | - | 0.88 | - |
| 17 | $^{126}$Sr | - | 1.19 | - | 11.60 | - | 0.302 | - | 1.25 | - | 2.13 | - |
| 16 | $^{124}$Sr | - | 1.39 | - | 11.56 | - | 0.306 | - | 0.80 | - | 2.93 | - |
| 15 | $^{122}$Sr | $^{122}$Sr | 1.62 | 8.63 | 11.49 | 11.59 | 0.311 | 0.311 | 3.60 | 2.64 | 6.53 | 2.64 |
| 14 | $^{121}$Y | $^{120}$Sr | 1.96 | 10.01 | 11.38 | 11.52 | 0.322 | 0.317 | 1.60 | 1.90 | 8.13 | 4.54 |
| 13 | $^{122}$Zr | $^{121}$Y | 2.17 | - | 11.31 | - | 0.328 | 0.322 | 1.48 | 1.10 | 9.61 | 5.64 |
| 12 | $^{124}$Mo | $^{122}$Zr | 2.59 | 10.57 | 11.20 | 11.39 | 0.339 | 0.328 | 3.87 | 2.20 | 13.48 | 7.84 |
| 11 | $^{80}$Ni | $^{124}$Mo | 3.19 | 11.44 | 11.01 | 11.24 | 0.350 | 0.339 | 4.95 | 4.10 | 18.43 | 11.94 |
| 10 | $^{78}$Ni | $^{126}$Ru | 3.80 | - | 10.84 | - | 0.359 | 0.349 | 1.74 | 1.05 | 20.17 | 12.99 |
| 9 | $^{76}$Ni | $^{78}$Ni | 4.27 | 13.76 | 10.74 | 10.88 | 0.368 | 0.359 | 4.92 | 7.00 | 25.09 | 19.99 |
| 8 | - | $^{80}$Zn | - | 14.74 | - | 10.58 | - | 0.375 | - | 5.10 | - | 25.09 |

\* *layer numbers start from the outer edge of the outer crust.*

Despite the importance of the NS crust described above, the equation of state of matter at a density above the nuclear one is also important for the stripping model. Two aspects can be considered here. First, the EoS affects the NS mass–radius curves and, hence, the parameters of the stripping process, in particular, its duration. Second, the composition of the NS core is currently unknown. It is important to emphasize here that, in contrast to the merger model, in the stripping one, not only the crust but the entire low-mass NS, including the core, experiences explosive decompression. The existence of exotic phases (for example, quarks, see, e.g., [28] and references therein) in the core can have intriguing consequences for the process of the LMNS explosion. This topic merits further investigation.

The article consists of two parts: in the first part, Section 2, we examine the influence of the NS EoS on the duration of the stable mass transfer (or the stripping time $t_{\mathrm{str}}$), the most important dynamical parameter of the model. A comparison of this parameter with the time delay ($t_{\mathrm{str}} \approx 1.7$ s) between the peak of the GW170817 signal and the detection of the GRB170817A by the FERMI and Integral satellites provides a new restriction on the parameters of nuclear matter.

In the second part, Section 3, the process of nucleosynthesis during the LMNS explosion is studied. First, we consider the differences in the r-process occurring in the outer crust related to the use of the different EoSs. Then, we examine the question of the uncertainties connected with the description of nucleosynthesis in the inner crust. The question of the influence of the nuclear data used (beta decay rates, mainly) on the results of the r-process is considered last. Our findings are presented in the conclusion section.

## 2. The Influence of the NS EoS on the Stripping Time $t_{\text{str}}$

### 2.1. The NS Inspiral Stage

Let us consider the last stages of the NS-NS binary system evolution using an analytical approach [1]. Two NSs with masses $M_1$ and $M_2$ ($M_1 \geq M_2$) rotate in a quasi-circular orbit with the orbital frequency

$$\Omega_{\text{orb}} = \sqrt{\frac{GM_{\text{tot}}}{a^3}}, \tag{1}$$

where $a$ is the distance between the components, and $M_{\text{tot}} = M_1 + M_2$ is the total mass of the system. The circular orbit assumption is justified by the population synthesis calculations [29] and the GW observations [30]. The orbital angular momentum $J_{\text{orb}}$ of the system in this case can be written as

$$J_{\text{orb}} = \frac{M_1 M_2}{M_{\text{tot}}} a^2 \Omega_{\text{orb}}. \tag{2}$$

The NSs approach each other due to the loss of the total angular momentum $J_{\text{tot}}$ of the system, which, in addition to the orbital angular momentum $J_{\text{orb}}$, includes the rotational (or spin) angular momenta of the components $J_1$ and $J_2$. The equation for changing the total angular momentum of the system has the form:

$$\dot{J}_{\text{GW}} = \dot{J}_{\text{orb}} + \dot{J}_1 + \dot{J}_2, \tag{3}$$

where $\dot{J}_{\text{GW}}$ is the rate of the angular momentum loss to emit GW, determined by the classical formula (e.g., [31]):

$$\dot{J}_{\text{GW}} = -\frac{32}{5} \frac{G}{c^5} \frac{M_1^2 M_2^2}{M_{\text{tot}}^2} a^4 \Omega_{\text{orb}}^5. \tag{4}$$

Let us assume that over millions of years of co-evolution, the NSs have been tidally synchronized (e.g., [32]). Then, their spin angular momenta before the beginning of the mass transfer are

$$J_{1,2} = I_{1,2}(M_{1,2}, J_{1,2})\Omega_{\text{orb}}, \tag{5}$$

where $I_{1,2}$ are the moments of inertia of the components. In the general case, the moment of inertia and the equatorial radius of a rotating NS depend on its mass and angular momentum. To calculate them, we use the approximate formulas from Appendix B of our paper [23]. Taking into account (5), it is easy to find the derivatives of the spin angular momenta with respect to time:

$$\dot{J}_{1,2} = \left[ I_{1,2}\dot{\Omega}_{\text{orb}} + \Omega_{\text{orb}}\dot{M}_{1,2}\left(\frac{\partial I_{1,2}}{\partial M_{1,2}}\right)_{J_{1,2}} \right] \beta_{1,2}, \tag{6}$$

where we introduce the notation $\beta_{1,2} = \left[1 - \Omega_{\text{orb}}\left(\frac{\partial I_{1,2}}{\partial J_{1,2}}\right)_{M_{1,2}}\right]^{-1}$. The system of Equations (1)–(6) describes the evolution of the NS-NS binary system before the beginning of the mass transfer.

### 2.2. The Stable Mass Transfer Stage

In the case of a sufficiently high asymmetry of the initial masses of the components (see [23] for the specific value), the LMNS with the radius $R_2$ at some moment first fills its Roche lobe with an effective radius $R_R$, i.e., $R_2 = R_R$. We parametrize the effective radius of the Roche lobe in accordance with [33]:

$$R_R = af(q'), \quad f(q') = \frac{0.49(q')^{2/3}}{0.6(q')^{2/3} + \ln\left[1 + (q')^{1/3}\right]}, \tag{7}$$

where $q' = M_2/M_1$ is the mass ratio of the components.

After the low-mass component fills its Roche lobe, the stable mass transfer onto the massive component through the inner Lagrangian point $L_1$ begins. At the same time, the orbital angular momentum $J_{\mathrm{orb}}$ of the system is partially transferred to the spin angular momentum $J_1$ of the accretor (see [34]):

$$\dot{J}_1 = -\dot{M}_2 \mathrm{j}(q, r_1) a^2 \Omega_{\mathrm{orb}}, \tag{8}$$

where j is the specific angular momentum of the accreting matter in orbital units, $q = M_2/M_{\mathrm{tot}}$ is the ratio of the donor mass to the total mass of the system, and $r_1$ is the dimensionless stopping radius of the accreting matter. During the stripping of the LMNS, two modes of accretion can take place [35]. In one case, the accretion stream hits the surface of the accretor with the equatorial radius $R_1$. If the minimum distance $R_{\mathrm{m}}$ for which the stream approaches the massive component turns out to be larger than the equatorial radius of the accretor $R_1$, then an accretion disk with outer radius $R_{\mathrm{d}}$ is formed. For various accretion modes (direct impact or disk accretion), the dimensionless stopping radius is:

$$r_1 = \begin{cases} R_1/a, \ R_1 \geqslant R_{\mathrm{m}}, \\ R_{\mathrm{d}}/a, \ R_1 < R_{\mathrm{m}}. \end{cases} \tag{9}$$

Approximations of $\mathrm{j}(q, r_1)$, $R_{\mathrm{m}}$, and $R_{\mathrm{d}}$ are given in [34].

Due to the accretion of matter, the asymmetry of the system increases and the components recede from each other. The mass transfer lasts on a relatively long time scale, determined by the rate of the orbital angular momentum loss emitted away by GW. If the total mass of the system is greater than the maximum NS mass, then at some moment the massive NS collapses into a BH. In this case, we assume the radius of the accretor to be $R_1 = 3R_g$, where $R_g$ is the Schwarzschild radius of the BH with mass $M_1$. For the rationale for this approach, see our paper [23].

The stable mass transfer continues until the size of the Roche lobe (7) grows faster than the radius of the LMNS $R_2$. This condition can be expressed as an inequality (see [22,23]):

$$\frac{d \ln R_2}{d \ln M_2} \geqslant \frac{d \ln f}{d \ln q} - 2 \frac{1 - 2q - \mathrm{j}(q, r_1) + \frac{\beta_2}{a^2} \left( \frac{\partial I_2}{\partial M_2} \right)_{J_2}}{1 - q - \frac{3\beta_2}{q} \frac{I_2}{M_2 a^2}}. \tag{10}$$

At some moment, the mass of the LMNS becomes so small that the mass transfer stability is violated. After that, the remnant $M_2 = M_{\mathrm{us}}$ (see Figure 3 below) is absorbed by $M_1$ on a fast, hydrodynamic time scale. When the low-mass component reaches the minimum NS mass $M_{\mathrm{min}}$, it loses its hydrodynamic stability and explodes [18].

*2.3. The NS EoS in the Low-Mass Region*

In our previous papers [22,23], the effects of accretion spin-up of the massive component, as well as its tidal and magneto-dipole spin-down, were investigated in detail. We have shown that accounting for accretion spin-up leads to a significant (by an order of magnitude) decrease in the stripping time $t_{\mathrm{str}}$, one of the most important parameters of the stripping mechanism, corresponding to the time delay between the loss of the GW signal and the GRB detection for GW170817-GRB170817A event [21]. The influence of the other two mentioned effects turns out to be insignificant.

In this article, we consider the contribution of the other important ingredient, the NS EoS. As was shown in [23], a specific type of EoS has a small impact on the position of the mass boundary between the merging and stripping scenarios. This is due to the fact that the derivative $\frac{d \ln R_2}{d \ln M_2}$ in the criterion (10) is almost equal to zero in the region of the moderate NS masses. But with decreasing of the LMNS mass, the contribution of this derivative increases (in absolute value), see Figure 2. Therefore, it is expected that the stripping time $t_{\mathrm{str}}$ should be sensitive to the NS EoS in the low-mass region.

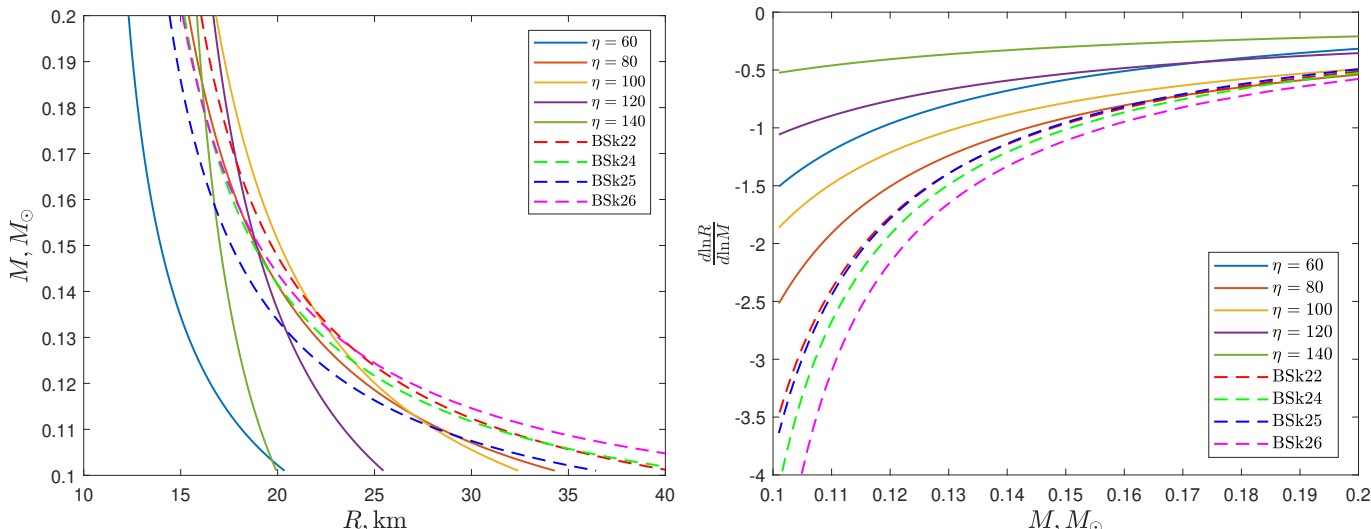

**Figure 2.** The NS mass–radius relations and its logarithmic derivative for different values of $\eta = (K_0 L^2)^{1/3}$ (in MeV) from [36] and the BSk-type EoS [26] in the low-mass region.

To investigate the sensitivity of the stripping time to the EoS in the low-mass region ($M < 0.4\,M_\odot$), we use the parametrization of the mass–radius relations from [36], shown in the left panel of Figure 2. Additionally, the dotted lines represent the curves corresponding to the latest versions of the BSk-type EoS [26]. The main parameter here is $\eta = (K_0 L^2)^{1/3}$, which is a combination of the so-called incompressibility of symmetric nuclear matter $K_0$ and the symmetry energy slope parameter $L = 3n_0(dS/dn_b)_{n_b=n_0}$, poorly determined from terrestrial experiments and astrophysical observations (see, e.g., [37,38] for a review). Recall that these parameters enter into the expansion of the nuclear EoS near the saturation density $n_0 \approx 0.16\,\text{fm}^{-3}$:

$$\omega = \omega_0 + \frac{K_0}{18 n_0^2}(n_b - n_0)^2 + \left[ S_0 + \frac{L}{3n_0}(n_b - n_0) \right]\alpha^2, \tag{11}$$

where $\omega_0 \approx -16$ MeV and $S_0 = S(n_0)$ are the saturation energy and the symmetry energy at the point $n_b = n_0$, and $\alpha \equiv (n_n - n_p)/n_b$ (all the symbols correspond to [36]).

How does the stripping time depend on the NS mass–radius relations (or the EoS)? This can be understood from the following qualitative considerations. With decreasing the LMNS mass, the radius of the flatter mass–radius configurations grows faster than the radius of the steeper ones. Therefore, the moment of the stability loss of the mass transfer, when the the Roche lobe size of the low-mass component grows slower than its radius, begins earlier for the flatter configurations. To confirm this argument, let us turn again to Figure 2. It can be seen from a comparison of the two panels that for the flatter mass–radius relations, the corresponding logarithmic derivative curves lie lower than for the steeper ones, so, according to our criterion (10), the stripping time is $t_{\text{str}}$ must be less for the flatter relations.

At the same time, as shown by our calculations, the last stages of the stripping process are much slower than the initial ones (see panel *d* in Figure 3 from [23]). This means that the stripping time is determined mainly by the last stages of the stable mass-transfer process. We also noticed that the NS-NS systems with the same total mass $M_{\text{tot}}$ but different initial mass ratios $q_2' < q_1'$ evolve through almost the same stages during the stripping process, starting from $q_2'$. In other words, the evolution of the NS-NS system weakly depends on its prehistory.

Using the results described above, we performed a series of calculations for the NS-NS system with initial masses $M_1 = 2.2\,M_\odot$ and $M_2 = 0.4\,M_\odot$ so that the total mass $M_{\text{tot}} = M_1 + M_2 = 2.6\,M_\odot$ agreed with the total mass of the GW170817 source [10]. We used the BSk22 EoS for the massive NS, and the parametrization $R_2 = R(M_2, \eta)$ — for

the LMNS. Accretion spin-up was taken into account in all calculations. Figure 3 shows the dependence of the stripping time $t_{str}$ and the mass of the low-mass component $M_{us}$ (unstable), at which the mass-transfer stability is lost, on the value of the nuclear parameter $\eta$. The horizontal dash-dotted line on the bottom panel corresponds to the minimum NS mass $M_{min} \approx 0.09\,M_\odot$ (see, e.g., [17]). We exclude the values of $\eta \lesssim 45$ MeV and $\eta \gtrsim 155$ MeV, which give non-physical results $M_{us} < M_{min}$. The DI region corresponds to the direct impact accretion of matter on the surface of the massive component, and DI+DF corresponds to the successive change of accretion modes and the accretion disk formation at the final stages of the stripping process. Crosses indicate calculations with the BSk EoS for the low-mass component. A comparison of Figure 3 with Figure 2 shows that, as we assumed, for the flatter mass–radius relations, the mass-transfer stability is lost for large $M_{us}$ and corresponds to small $t_{str}$.

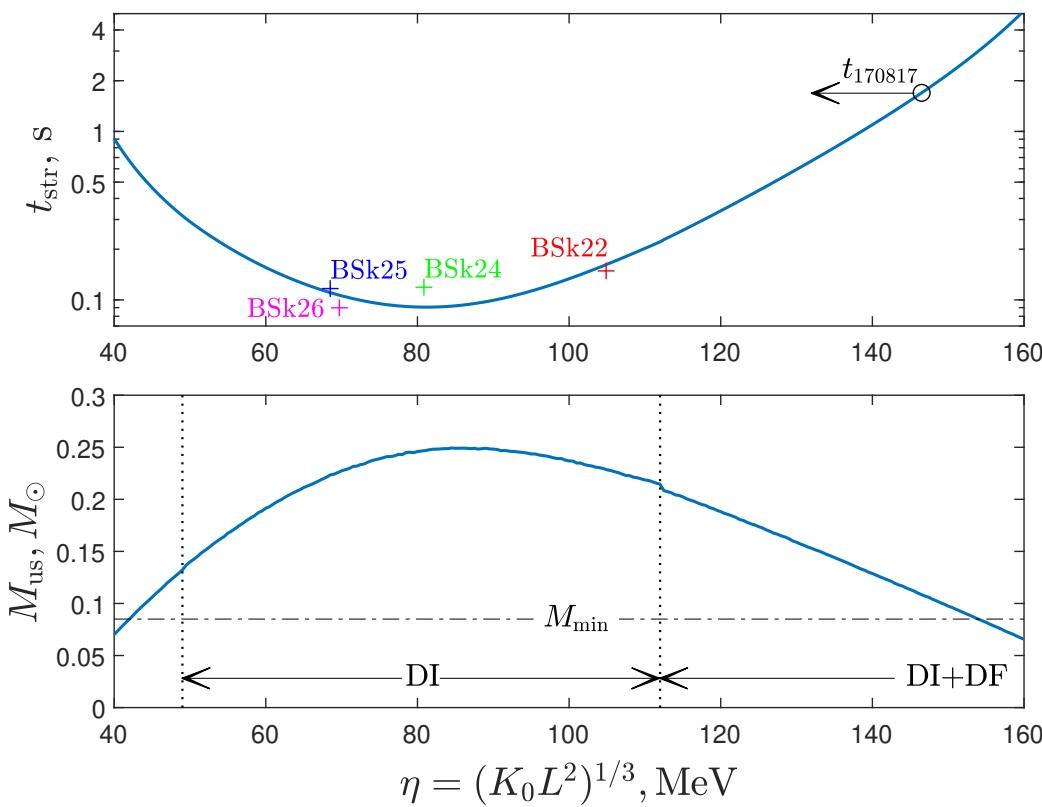

**Figure 3.** The top and bottom panels show the dependence of the stripping time $t_{str}$ and the LMNS mass $M_{us}$, at which the mass-transfer stability is lost, on the parameter $\eta$. Crosses indicate calculations for the BSk EoS. See text for details.

### 2.4. The Nuclear Parameters and the Stripping Time

The identification of the GW170817-GRB170817A event in the framework of the stripping model [21,22] allows us to impose an important limitation on the nuclear EoS parameters near the saturation density point from (11). Let us look at Figure 3 again. The circle on the top panel corresponds to the observed stripping time $t_{str} \approx 1.7$ s, corresponding to the time delay between the loss of the GW170817 signal by the LIGO-Virgo GW interferometers and the registration of the GRB170817A [7,8]. A comparison of the calculated and observed values of $t_{str}$ allows us to find the nuclear parameter $\eta = 146.5$ MeV. The value of $\eta$ obtained in this way is the upper bound because in our calculations with the formula (3) we neglected the effect of tidal spin-down of the massive component, which slightly increases the duration of the stable mass transfer (see [23]). As discussed above, we perform calculations with initial masses $M_1 = 2.2\,M_\odot$ and $M_2 = 0.4\,M_\odot$. At the same time, the generation of the GRB during the explosion of the minimum NS mass after the loss of

the mass-transfer stability (at $M_2 = M_{us}$) should also take some extra time [27]. All these processes should increase the stripping time and correspondingly reduce the real value of $\eta = (K_0 L^2)^{1/3}$. The constraint on the range of parameters $L - K_0$ is shown in Figure 4. The yellow area illustrates the limitations according to the PREX-II experiment [39]. In this important experiment, the neutron skin thickness of $^{208}$Pb, which correlates with the NS radius and the symmetry energy slope parameter $L$ [40], was determined for the first time in a model-independent way. The blue area (Astro) corresponds to the results of processing various astrophysical observations related to determining the NS masses and radii [41]. The blue ellipse combines astrophysical observations with the constraints on the NS EoS from the chiral effective field theory, $\chi$EFT (see also [42] for a review). The plus signs denote the values of $L$ and $K_0$ for the BSk-type EoS [26].

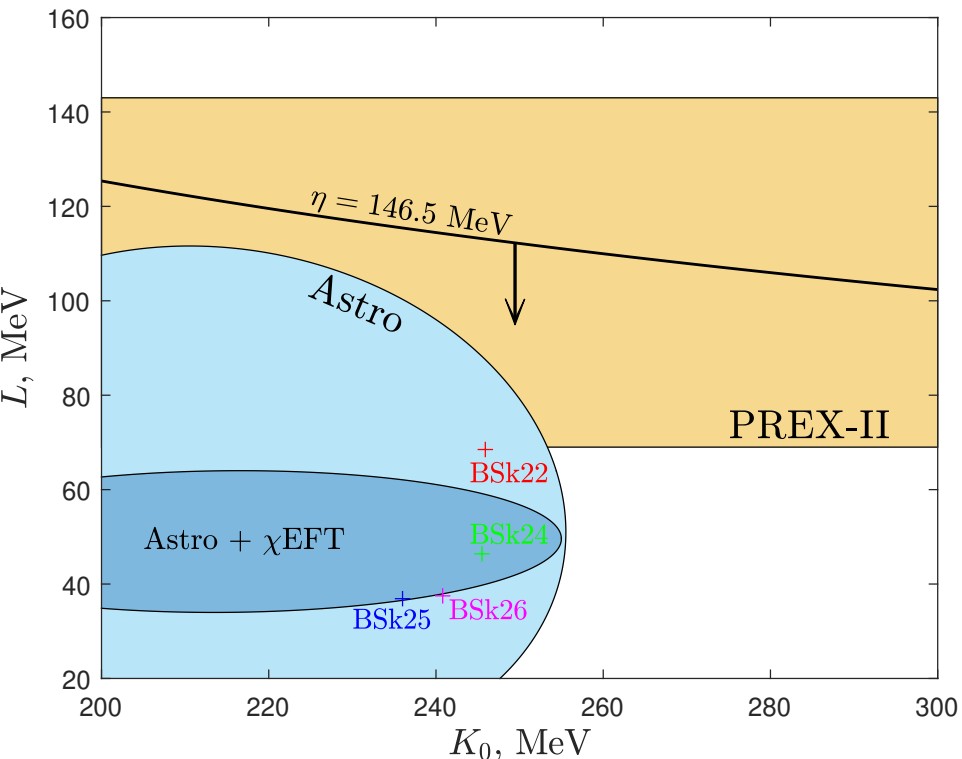

**Figure 4.** Constraints on the parameters of the nuclear EoS obtained from the PREX-II experiment and the combined astrophysical observations. The black line corresponds to the upper bound obtained from comparing the calculated stripping time with the GW170817-GRB170817A time delay. See the text for details.

Having the generally accepted value of the incompressibility of symmetric nuclear matter $K_0 = 240$ MeV (see, e.g., [43]), we obtain $L < 114.5$ MeV (see Figure 5). The colored symbols (together with the corresponding error bars) represent the results of the Bayesian analysis of various terrestrial experiments and astronomical observations. The orange rhombus denotes the result of processing model-dependent measurements of the neutron skin thickness of tin isotopes (Sn-isotopes) [44]. The green square corresponds to the limitation from observations of low-mass X-ray binaries with the Chandra and XMM-Newton telescopes [45], and the purple five-pointed star is obtained from the analysis of the GW170817 signal, combined with data from various experiments to measure the neutron skin thickness of lead [46]. Figure 5 clearly shows a significant discrepancy between the results of the mentioned PREX-II [39] experiment (the four-pointed asterisk) and the entire set of astrophysical observations [41] (the blue triangle), as well as the other terrestrial experiments. However, we note that an alternative analysis [47] of PREX-II gives significantly lower value of $L$ (the blue cross), which appears to be consistent with

other estimates. It can be seen that our constraint on $L$ (the burgundy circle) agrees with all the data presented. The subsequent consideration of the general relativity effects and non-conservative mass transfer discussed in [23] will make it possible to refine this estimate.

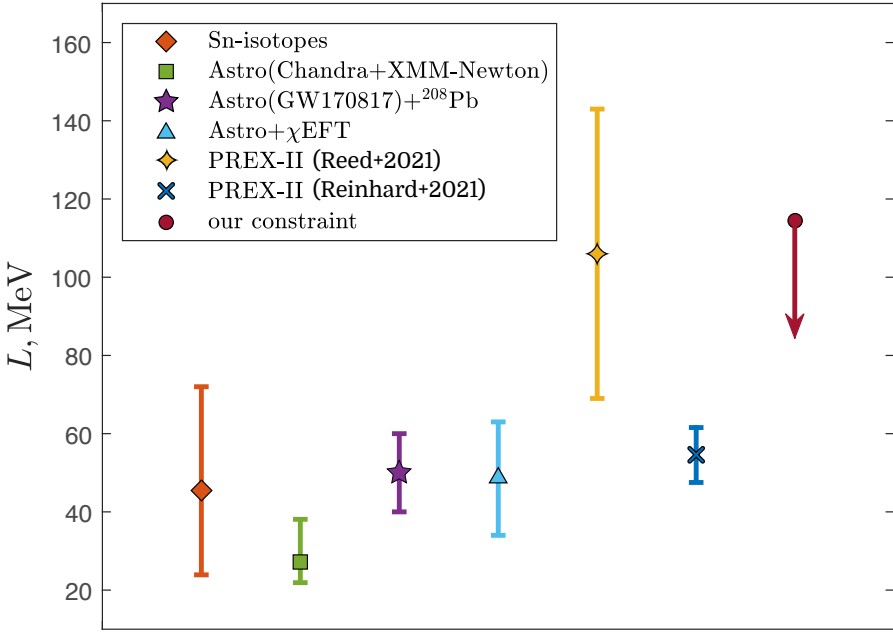

**Figure 5.** Limitations of the symmetry energy slope parameter from various experiments and astrophysical observations. See text for details.

## 3. r-Process during the LMNS Explosion

### 3.1. On the Influence of the EoS on the Results of Nucleosynthesis

The EoS is also necessary for modeling the explosion of an NS and obtaining realistic trajectories for subsequent calculations of nucleosynthesis. Today, there are quite a lot of different EoSs [36], and all of them give different predictions regarding, for example, the characteristics of an NS, such as its mass, radius, and structure. Therefore, it is extremely important to understand how this ambiguity in the choice of the EoS can affect the parameters of the NS explosion and the results of the accompanying nucleosynthesis.

When considering the impact of nuclear data on nucleosynthesis during the explosion of the minimum NS mass [19,20], we decided to compare the results obtained using different EoSs. In the first stage, we examine the results of formation of new elements only in the outer crust [48].

We have considered two variants with different approximations of the NS EoS, BSk22, and BSk25 [26], leading to maximum differences in the EoS, especially for relatively low densities. These EoSs are based on a family of mass models derived using the Hartree-Fock-Bogolyubov [49] method, fitted using known nuclear masses (AME2012) [50] and based on realistic nuclear force expressions.

As it turned out, the use of different EoSs leads to different dynamics of expansion of the shells of the LMNS [19,27] and, as a result, to different duration of nucleosynthesis and the formation of different sets of chemical elements. Thus, the shock wave propagation velocity for BSk25 turns out to be approximately twice as high, and at the peak, the temperature value is many times greater than in calculations with the BSk22 model (see Table 1).

Having the data for the layers of the outer crust in two variants of calculations (Table 1) for different EoSs, we obtained integral curves for the abundance of elements $Y(A)$ (Figure 6) formed during the expansion of this region of the star. The indicated abundances were obtained for the same mass of matter in both series of calculations: for the sum of layers with a mass of $M \sim 0.002509 \, M_{\odot}$ with a total thickness of these layers of $\sim 8.0$ km. In

the BSk22 model (the outer crust consists of 18 layers of different composition), the results of layers from 9 to 18 were summed, and in the calculations with BSk25 (the outer crust has 15 layers), layers from 8 to 15 were taken into account, see Table 1 and Figure 1.

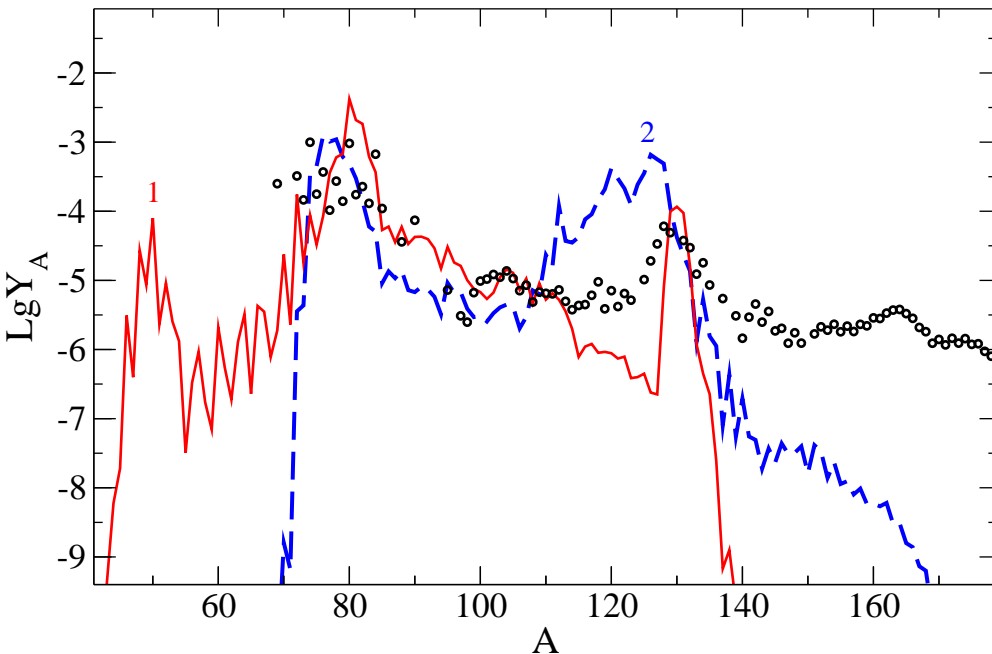

**Figure 6.** Integral curve $Y(A)$ of the results of nucleosynthesis in the outer crust using the BSk22 (curve 2, blue) and BSk25 (curve 1, red). The layers of the outer crust are taken into account, starting from No. 9 (see Table 1), having a total mass of $M \sim 0.0025 M_\odot$. Solar abundance is shown by dots.

As a result of nucleosynthesis, elements in the outer crust are mainly formed in the range of mass numbers from $A = 80$ to $A = 130$ for both considered EoSs. As can be seen, the integral results on the abundance $Y(A)$ obtained by modeling the explosion of the LMNS [19,20] using different EoSs (BSk22, BSk25) differ markedly, especially in the region of the cadmium peak.

The difference in the results is mainly due to different temperature dynamics along the considered trajectories when using different EoSs. So, in calculations with BSk25, the shock wave heats the matter to values 3–4 times higher than when using BSk22 (see Table 1), and as a result of photo-nuclear reactions, quite a lot of nuclei with $A < 70$ appear. The key factor is the magnitude of the temperature jump when the shock wave reaches the layer (see the values of $T_9^{\max}(t)$ in Table 1). When using the BSk25 model, the maximum temperature is several times higher than BSk22, which leads to a much more intense photo-dissociation of most of the heavy nuclei formed in the weak r-process and to a decrease in their atomic and mass numbers. After the passage of the shock wave, the density of free neutrons $n_n$ in most layers is small for the resumption of the r-process, and the final elemental composition of the layer material is mainly determined by the occurrence of explosive nucleosynthesis. In a series of calculations with BSk22, the nuclei formed before the arrival of the shock wave were not so actively destroyed since the layers are not heated by the shock wave so strongly.

Summarizing, we note that our calculations show a noticeably greater dependence on the EoS than, for example, in [51]. Perhaps the reason is that here we only consider nucleosynthesis in layers of the outer crust for a limited range of values of the ratio of the number of neutrons to the number of seed nuclei, in which $Y_e > 0.3$.

### 3.2. Nucleosynthesis in the Inner Crust

The calculation of nucleosynthesis in the inner crust of NS has much larger uncertainties than in the outer one and was considered by us on the explosive trajectory obtained

using the BSk25 EoS. The structure of the inner crust for this EoS is shown in Figure 7 (see also Figure 1). The inner crust consists of neutron-rich hyper-nuclei (clusters) immersed in a sea of free neutrons [17].

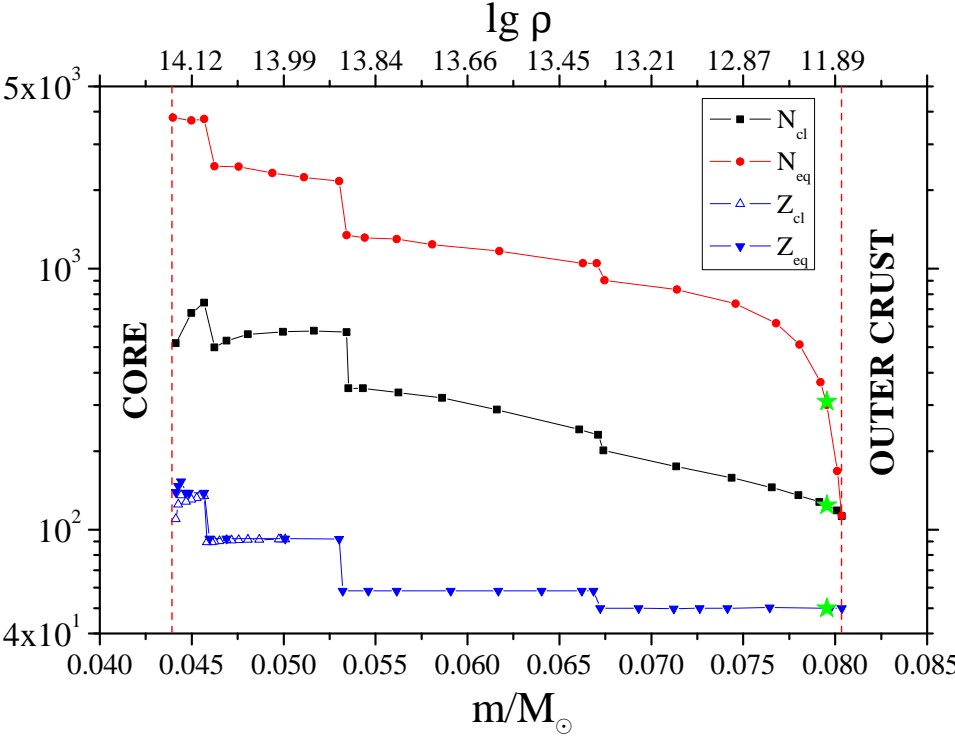

**Figure 7.** Composition of the inner crust for BSk25 as a function of the Lagrangian variable (mass) *m*. The corresponding density values are given on the upper axis. See text for details.

In the figure, the values of the following quantities are given as functions of the mass coordinate *m* in the inner crust of the NS. The blue triangles show $Z_{cl}$, the cluster charge, and $Z_{eq}$, the total charge of the Wigner–Seitz cell, coinciding with $Z_{cl}$ almost everywhere, except for the inner boundary of the crust (near the NS core). The black squares show $N_{cl}$, the number of neutrons bound in the cluster. The red circles are the total number of neutrons in cell $N_{eq}$, both free and bound. The corresponding density values are given on the upper axis.

The main issue for calculating nucleosynthesis in the inner crust is how exactly the decompression takes place. During the expansion of the crust matter and the rapid decompression, the clusters must dissociate, eventually giving rise to seed nuclei to start calculating the r-process. But how does this process take place? Does the cluster evaporate "extra" neutrons? Do they have time to undergo beta decay, thereby reducing the neutron excess and increasing the charge number of seed nuclei?

To clarify these issues, the outermost zone of the inner crust was considered. Firstly, in contrast to the outer crust, conditions in this zone are created for an intense r-process. Secondly, this zone is the simplest in modeling physical processes. In Figure 7, the position of this zone is shown by the green stars. Two models of transformation of the initial nuclei of the crust during decompression during the explosion into seed nuclei for the r-process are given in Table 2. These models implement two limiting cases of decompression: (1) neutrons evaporate during decompression until the neutron binding energy $S_n$ becomes positive, or (2) a chain of beta decays occurs until the same condition is met, $S_n > 0$. The mono-nuclear composition of the considered zone is also marked with the green star in Figure 8. This figure is a $N-Z$ nuclide chart, on which the area of bound nuclei taken into account in our calculations of nucleosynthesis is shaded in black. As can be seen, the parameters of the cluster under consideration lie outside the region of known nuclei, and

in the process of decompression, the initial nucleus is transformed into the region of bound nuclei between the boundaries of neutron and proton stability.

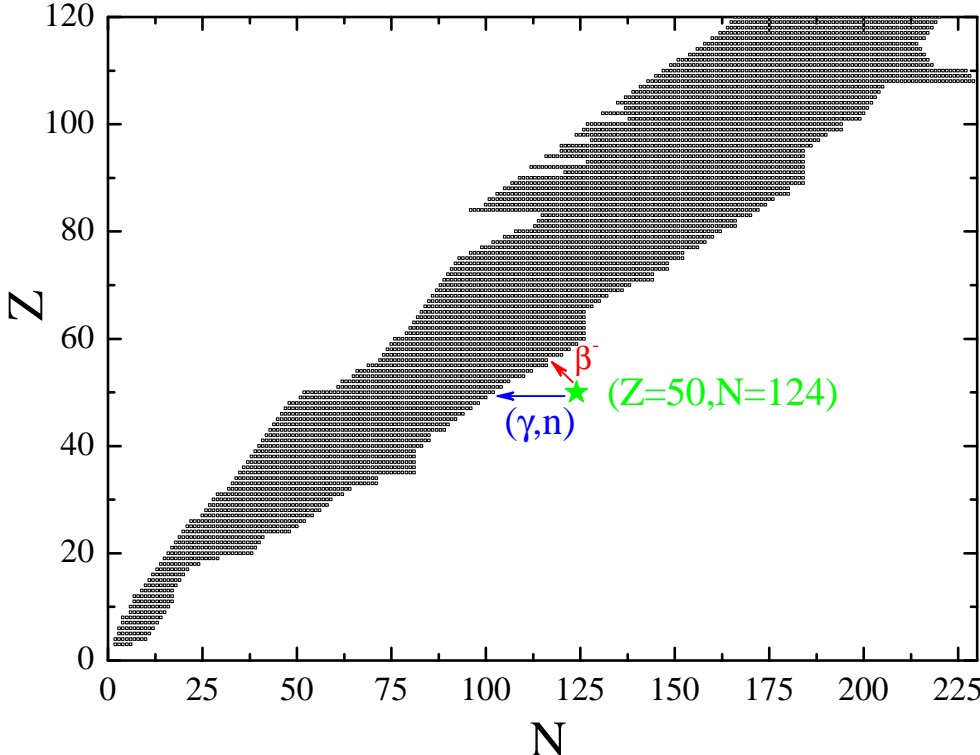

**Figure 8.** $N-Z$ chart of nuclei included in our calculations. The position of the neutron-rich cluster is indicated by a green asterisk. The arrows indicate the direction of dissociation of this cluster in two limiting cases.

**Table 2.** Seed nuclei formation in two decompression models of layer 16 (BSk25).

| Model | Before Decompression | | | | After Decompression | | | | |
|---|---|---|---|---|---|---|---|---|---|
| | $Z$ | $A$ | $N_{\text{free}}$ | Evaporated n | $Z_{\text{seed}}$ | $A_{\text{seed}}$ | $Y_{\text{seed}}$ | $Y_{\text{n}}$ | $Y_{\text{e}}$ |
| $(\gamma, n)$ | 50 | 174 | 186 | 22 | 50 | 152 | 0.00278 | 0.578 | 0.139 |
| $\beta^-$ | 50 | 174 | 186 | 0 | 54 | 174 | 0.00278 | 0.517 | 0.150 |

In Figure 8, the arrows indicate the paths of transformation of the initial nucleus, when the cluster either evaporates "extra" neutrons during an explosive decrease in density (blue arrow to the left with the caption $(\gamma, n)$) or undergoes a chain of beta decays up to reaching the boundary of known nuclei (red arrow to the left-up with the caption $\beta^-$).

Table 2 lists the parameters of two extreme cases that implement these features. The original neutron-rich $^{174}50$ nucleus, as a result of decompression, is transformed or due to the evaporation of neutrons into the $^{152}50$ nucleus (evaporated neutrons are added to free $N_{\text{free}}$), or through only beta-decays into a nucleus $^{174}54$ at the boundary of bound nuclei.

Nucleosynthesis calculations for these two cases are shown in Figure 9. They demonstrate a weak dependence of the final abundance $Y(A)$ on the decompression options considered, which is apparently due to the strong influence of the "fission cycling" [52]). This result inspires a certain optimism and allows us to hope that the yield of nucleosynthesis in the inner crust will be resistant to the details of the decompression of neutron-rich clusters, a consistent model of which, however, still needs to be developed. For deeper layers of the inner crust, the difference between these two limiting cases

considered should be even smaller due to the increasing influence of "fission cycling" in a more neutron-rich environment.

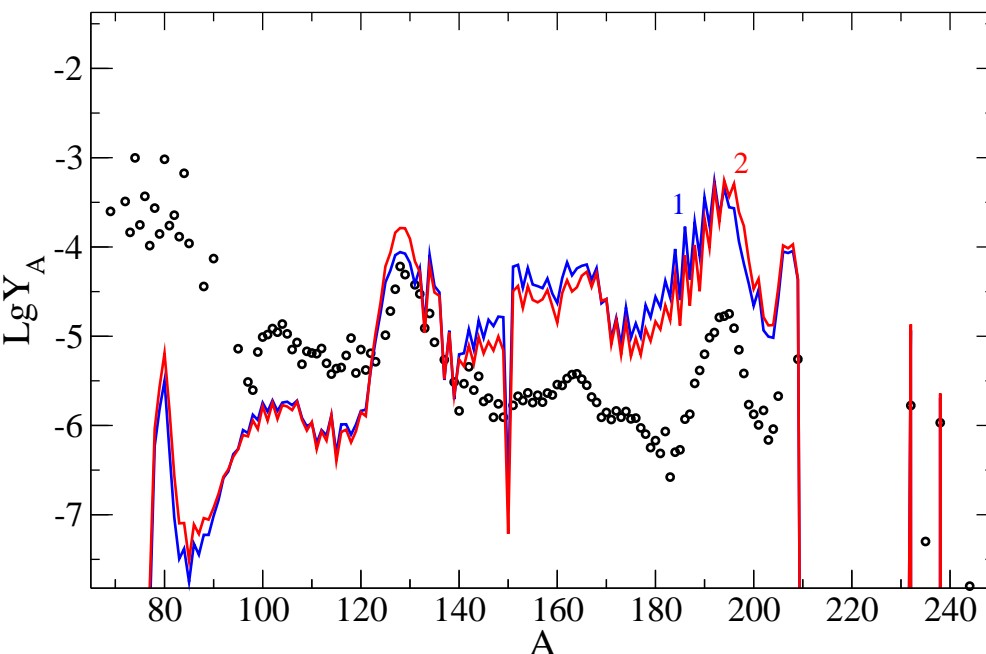

**Figure 9.** The result of nucleosynthesis $Y(A)$ depending on the limit decompression options: only evaporation of neutrons $(\gamma, n)$ (curve 1, blue) or only beta decays $\beta^-$ (curve 2, red). Dots are the solar abundance.

### 3.3. The Influence of Beta Decay Rates

The amount of heavy elements synthesized in the r-process depends both on the duration of the neutron exposure and on the speed of the nucleosynthesis wave. And the rate of nucleosynthesis depends on the conditions and region of the r-process, determined by the astrophysical scenario, and on the rate of beta-decay of the nuclei involved in the r-process. As the nucleosynthesis wave accelerates or slows down in the region of heavier nuclei, the r-process trajectory on the nuclei chart changes [53], as well as the position of the third peak on the heavy-element abundance curve [54], which indicates the multifaceted influence of the model beta decay to nucleosynthesis.

In recent years, beta decay models have been intensively developed, which has led to the emergence of new global calculations of these important characteristics [55,56], expanding the possibility, on the one hand, to study the stability of the results of modeling the nucleosynthesis process to nuclear data and, on the other hand, to try to evaluate, with the help of observations, the reliability of one or another nuclear-physical model for predicting the characteristics of exotic nuclei.

Calculations based on the density functional DF3 [57,58], which describe the characteristics of neutron-rich spherical nuclei, have so far been carried out only for a part of the nuclei involved in the r-process. The existing number of data are still insufficient for their use in full-fledged calculations of the r-process. Here, as an illustration of the influence of the beta decay model, we used beta-decay rates calculations based on the QRPA [59] and global calculations based on the finite amplitude method (FAM) [56].

Figure 10 shows the results of the calculation of nucleosynthesis in the r-process along the trajectory of the expansion of matter in the outer zone of the inner crust (see Figure 7), using beta decay rates derived from these two different approaches.

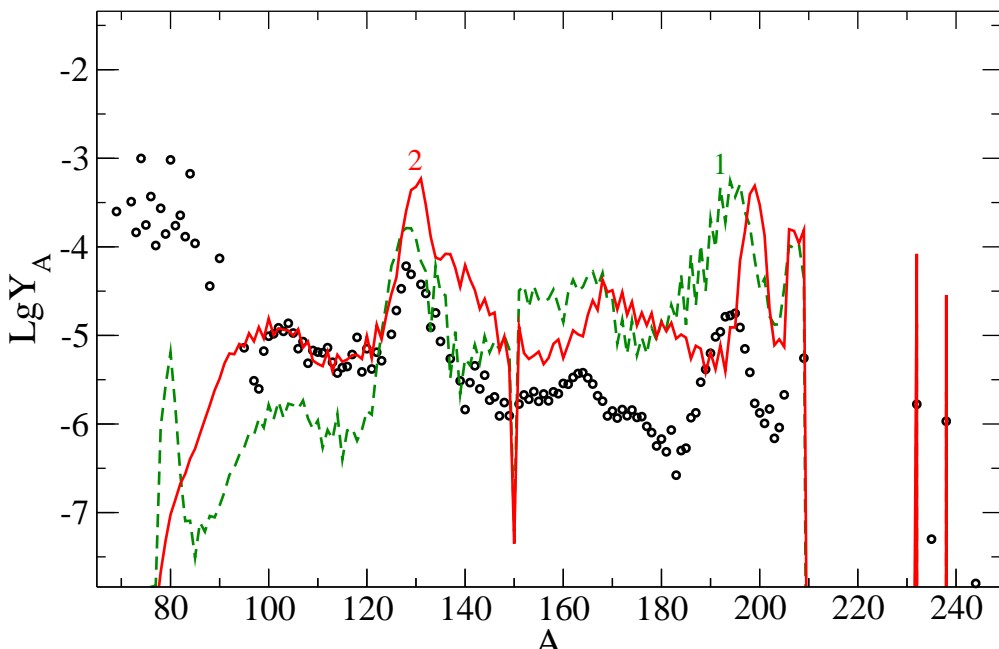

**Figure 10.** Dependence of the abundance of heavy nuclei for the outer layer of the inner crust when using different beta decay rates in the calculations: curve 1 (green) is for beta decay rates obtained from the drop model (FRDM) [59], and curve 2 (red) is based on the finite amplitude method (FAM) [56].

The influence of the model predicting the rate of beta-decay on the results of nucleosynthesis in the stripping scenario under consideration, although less than in the [60] merging scenario, is still significant and requires additional research. Of particular interest is the question of the reason for the shift of the platinum peak with respect to observations (points in Figure 10), which has been repeatedly discussed in the literature (see, for example, [61]) and is certainly important for understanding the process of its formation.

## 4. Discussion and Conclusions

In this paper, we considered the question of how the nuclear data used can influence the predictions of the stripping model for short GRBs. An essential ingredient of this model is the explosion of the minimum mass NS, during which gamma radiation is generated. Due to the specific structure of the LMNS, the problem of the properties of the NS crust, which here extends for hundreds of kilometers, plays an important role .

In the first part of the present work, we have investigated the influence of the EoS parameters in the region of low densities on the stripping time $t_{str}$, the duration of the stable mass transfer of matter in the stripping mechanism. In the pioneering work [1] for the moderate values of the masses of the components, the stripping time $t_{str} \approx 1.7$ s was obtained. Exactly the same value was recorded decades later as the time delay between the detection of the GW signal GW170817 and registration of gamma radiation by the FERMI and Integral satellites from GRB170817A. This remarkable coincidence have served us (among other things, see [21]) as a strong indication in favor of interpreting the GW170817-GRB170817A event as the result of the stripping rather than the merging mechanism.

However, in the subsequent work [23] we came to the disappointing conclusion that taking into account the accretion spin-up of the massive component during the mass transfer leads to a significant (by an order of magnitude) decrease in the stripping time. In this work, we have shown that the significant uncertainties in the NS EoS [36], especially in the low-mass region, can resolve this contradiction. Moreover, fixing the value $t_{str} = 1.7$ s (and other system parameters) of the GW170817-GRB170817A event makes it possible to constrain the parameters of nuclear matter, mainly the symmetry energy slope parameter $L < 114.5$ MeV.

The subsequent accounting for the general relativity effects, non-conservative mass transfer, etc., in our analytical calculations will make it possible to refine the obtained constraint.

The second part of this work is devoted to the influence of nuclear data on the nucleosynthesis process that accompanies the LMNS explosion [19,20]. We first showed that the final abundances of the elements is sensitive to the EoS used. This conclusion echoes the conclusions of the first part of our work. This is explained by the differences in the properties of matter heated by a shock wave passing through and cumulating in the region of the LMNS crust [27]. Here, we have investigated the influence of the EoS on the r-process precisely through the changed thermodynamic conditions in matter.

Then, we examine nucleosynthesis in the inner NS crust. There are many more uncertainties here; in particular, the nuclear composition of matter is much more complicated than a simple sequence of mono-layers of nuclei, as in the outer crust. However, as we have shown, the predictions of the nucleosynthesis model here are much clearer and less sensitive to the present uncertainties. This is explained by the significant influence of the "fission-cycling" effect [52], which causes the system to "forget" initial conditions very quickly, and fine details are blurred over wide distributions of a huge number of participating nuclei, repeatedly passing through the cycle: neutron capture—beta decay—fission.

In the final section, the example of nucleosynthesis in one of the inner crust shells has revealed a rather strong dependence of the results on the beta decay model. This is the basis for further study in order to assess the impact of nuclear data on the final integral results of the entire ejected matter, for which it is necessary to resolve some of the issues discussed above. Therefore, the discussion, in particular, of the dependence of the platinum peak position, which was considered earlier [61], is more appropriate to consider with the results of nucleosynthesis for the matter of the entire crust.

It should be specially noted that the dynamics of changing conditions along the trajectories are very different from the dynamics in the merging scenarios of the NSs with approximately equal masses, finally forming a supramassive NS. However, the resulting dependencies $Y(A)$ for both scenarios are similar; therefore, the stripping model should also be added to the list of the most probable scenarios for the r-process nucleosynthesis (see, e.g., [62]).

**Author Contributions:** Conceptualization: A.Y., N.K. and I.P.; writing—original draft preparation: A.Y., N.K. and I.P.; writing—review and editing: N.K. and I.P.; $t_{str}$ bounds: A.Y. and N.K.; and nucleosynthesis calculations: I.P. and A.I. All of the authors have read and agreed to the published version of the manuscript.

**Funding:** The authors are grateful to the RSF 21-12-00061 grant for support.

**Data Availability Statement:** Data generated from computations are reported in the body of the paper. Additional data can be made available upon reasonable request.

**Conflicts of Interest:** The authors declare no conflicts of interest.

## Abbreviations

The following abbreviations are used in this manuscript:

| | |
|---|---|
| GRB | Gamma-ray burst |
| GW | Gravitational wave |
| NS | Neutron star |
| LMNS | Low-mass NS |
| BH | Black hole |
| EoS | Equation of state |

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
