# Peer review of "Stripping Model for Short GRBs: The Impact of Nuclear Data"

_2571-712X, doi:10.3390/particles6030050_

Round 1
Reviewer 1 Report
This paper studies the effects of the equation of state (EoS) on the various nuclear processes that take place in the crusts of neutron stars during the merger process. The authors consider an alternative to the standard scenario of the merger, where the crust of the lighter star is pealed off by the binary interaction and it explodes prior to the formation of the hypermassive remnant of the merger. Limits are set on the characteristics of nuclear matter such as the slope of symmetry energy. The paper is well-written and the physics is accurately described. It can be recommended for publication in Particles after a minor revision according to the points below.
1. There is indeed an absolute minimum mass that follows from the solution of the Oppenheimer-Volkoff equations which is quoted as 0.085 solar mass. However, it is known that it is impossible to produce low-mass stars in supernova explosions, the lower limit being around 1.17 solar mass. So, the existence of very low-mass objects is questionable as has been discussed recently in the literature in the context of CCO HESS J1731-347.
2. As an update to Ref. 34, I would recommend citing also arXiv:2301.03666 by the same author.
3. An alternative analysis of PREX-II gives more reasonable values than the Reed et al one. See Phys. Rev. Lett. 127, 232501.
4. The core of a neutron star may contain not only nucleons but also heavy baryons and quarks. If possible provide a few sentences on how this will change the dependence of physics on the EoS and add a few references to reviews on the topic(s).
The English language is good.
Author Response
1. We agree with a respected reviewer that "It is impossible to produce Low-
Mass Stars in Supernova Explosions". Neutron star of a small mass
can be obtained only as a result of the mass exchange in a double system,
starting with a completely moderate mass value. We added the following
paragraph to the text (Introduction, page 2):
An important issue is the conditions under which the mechanism of stripping rather than merging is implemented. Our preliminary calculations \cite{KramarevYudin2023} show that this condition weakly depends on the total mass of the system, and is determined mainly by the mass ratio of the components. The stripping model is realized at $M_2/M_1 {\lesssim} 0.8$, so about a quarter of the observed galactic NS-NS binaries \cite{Farrow2019} with known masses must finish their evolution in accordance with this scenario. In this context, it is also interesting to mention the recent discovery \cite{Doroshenko2022} of very low-mass NS ($0.77M_\odot$ approximately). In any case, this issue requires further careful study.
2. Done!
3. We included this result both into the Fig. 5 and in the discussion around
it.
4. We added the following paragraph to Introduction, page 3:
Despite the importance of the NS crust described above, the equation of state of matter at a density above the nuclear one is also important for the stripping model. Two aspects can be considered here. First, the EoS affects the NS mass-radius curves and, hence, the parameters of the stripping process, in particular, its duration. Second, the composition of the NS core is currently unknown. It is important to emphasize here that, in contrast to the merger model, in the stripping one, not only the crust, but the entire low-mass NS, including the core, experiences explosive decompression. The existence of exotic phases (for example quarks, see e.g. \cite{Blaschke2022} and references therein) in the core can have intriguing consequences for the process of the LMNS explosion. This topic merit further investigation.
Reviewer 2 Report
This paper calculates the effects of undercertainties in nuclear physics data on the "stripping model" of short gamma ray bursts. As better data become available, the evolution of these systems will be better understood.
One weakness should be acknowledged: The stripping model assumes one neutron star substantially less massive than the other in a double neutron star binary. This may not be achievable: although stable neutron star structures may be calculated with masses as low as 0.085 M_\odot in the authors' preferred EOS, it is not known how stellar evolution may make neutron stars will mass less than about 1.2 M_\odot (the Chandrasekher mass of baryons, allowing for loss of the neutron star binding energy). The maximum possible neutron star mass is about 2.2 M_\odot, meaning the mass ratio cannot be less than 0.55.
Minor language issue: the authors use "incoming" when they mean "forthcoming" (will be known in the future).
Author Response
1. We agree that it is impossible to produce Low-Mass NS directly
in Supernova explosion. Neutron star of a small mass
can be obtained only as a result of the mass exchange in a double system,
starting with a completely moderate mass value. We added the following
paragraph to the text (Introduction, page 2):
An important issue is the conditions under which the mechanism of stripping rather than merging is implemented. Our preliminary calculations \cite{KramarevYudin2023} show that this condition weakly depends on the total mass of the system, and is determined mainly by the mass ratio of the components. The stripping model is realized at $M_2/M_1 {\lesssim} 0.8$, so about a quarter of the observed galactic NS-NS binaries \cite{Farrow2019} with known masses must finish their evolution in accordance with this scenario. In this context, it is also interesting to mention the recent discovery \cite{Doroshenko2022} of very low-mass NS ($0.77M_\odot$ approximately). In any case, this issue requires further careful study.
2. Corrected!
Reviewer 3 Report
Dear Editor.
Thank you so much for considering me as a reviewer of the manuscript "Stripping Model for Short GRBs: the impact of nuclear data". They discussed the impact of incoming nuclear data on the predictions of the neutron star stripping model for short gamma-ray bursts. They collected theoretical background and some various experiments and astrophysical observations in their manuscript which was very instructive to me and well readable.
After reading their manuscript and checking its structure of it, I'm so happy to inform you I recommend the present manuscript for publishing in this journal. I hope for more success for the authors.
Best regards.
Author Response
Thank you very much for your kind words!